# Control Strategies of Plastic Biodegradation through Adjusting Additives Ratios Using In Silico Approaches Associated with Proportional Factorial Experimental Design

**DOI:** 10.3390/ijerph19095670

**Published:** 2022-05-06

**Authors:** Haigang Zhang, Yilin Hou, Wenjin Zhao, Hui Na

**Affiliations:** 1Alan G. MacDiarmid Institute, Jilin University, Changchun 130012, China; zhanghg18@mails.jlu.edu.cn (H.Z.); huina@jlu.edu.cn (H.N.); 2Xi’an Boiler & Environmental Protection Engineering Co., Ltd., Xi’an 710054, China; yl_hou@cjhb.chng.com.cn; 3Huaneng Yangtze Environmental Technology Co., Ltd., Beijing 100031, China; 4College of New Energy and Environment, Jilin University, Changchun 130012, China

**Keywords:** polystyrene plastic, biodegradation, additive combination, factorial analysis, amino acid analysis

## Abstract

Plastics, as a polymer material, have long been a source of environmental concern. This paper uses polystyrene plastics as the research object, and the relative contribution of each component of plastic additives to plastic degradation is screened using the molecular dynamics method. The factorial experimental design method is combined with molecular dynamics simulation to adjust the additive composition scheme, analyze the mechanism of interaction between the additive components, and select the plastic additive combination that is most readily absorbed and degraded by microorganisms. Seven different types of plastic additives, including plasticizers, antioxidants, light and heat stabilizers, flame retardants, lubricants, and fillers, are chosen as external stimuli affecting the biodegradability of plastics. Using molecular dynamics simulation technology, it is demonstrated that plastic additives can promote the biodegradability of plastics. The factorial experimental design analysis revealed that all plastic additives can promote plastic biodegradation and plasticizer is the most favorable factor affecting plastic degradation, that hydrophobicity interactions are the primary reason for enhancing plastic degradation, and that screening No. 116–45 (plasticizer A, light stabilizer C, flame retardant E) is the most advantageous combination of biodegradable plastic additives. The plastic biodegradation effect regulation scheme proposed in this study is based on optimizing the proportion of additive components. To continue research on aquatic biodegradable plastics, the optimal combination of plastic components that can be absorbed and degraded by microorganisms is recommended.

## 1. Introduction

Plastic, a type of polymeric material obtained through the addition or condensation polymerization of a resin monomer as a raw material [1], is widely used in all facets of life and production due to its affordability and durability [2]. Nowadays, annual plastic production can reach 350 million tons [3] and the average person discards 52 kg of plastic each year [4]. There are numerous types of plastics, including polyethylene (PE), polypropylene (PP), polystyrene (PS), poly (ethylene terephthalate) (PET) and polyvinyl chloride (PVC), among others [5], that can cause loss during marine transportation, and their microplastic particles enter the ocean as a result of randomly discarded plastic waste or factory discharge channels [6,7].PS is widely consumed, and it is one of the most frequently monitored types of plastic fragments in the environment [8]. Numerous researchers have examined the toxicity of PS plastics to marine organisms. Wan et al. [9] discovered that PS plastic could alter the metabolic spectrum of juvenile zebrafish and impair glucose, lipid, and energy metabolism. Sussarellu et al. [10] discovered that oysters exposed to PS plastic experienced significant decreases in oocyte number, diameter, and sperm velocity of 38, 5 and 23%, respectively, and in offspring number and developmental velocity of 41 and 18%, respectively, confirming that exposure to PS microplastics causes reproductive disturbances in oysters and has serious consequences for offspring growth and development. Yu et al. [11] exposed *Cryptomeria hidradiens* to varying concentrations of PS microplastics to determine the toxicity and mechanism of exposure. They discovered that the degree of oxidative stress and intestinal damage were significantly correlated with the level of PS microplastic exposure. It is hypothesized that PS microplastics cause toxicity through oxidative stress and intestinal injury. As a result, consideration should be given to how to effectively manage PS plastics and mitigate their impact on the marine ecosystem.

Along with basic polymeric monomers, plastics contain a variety of plastic additives that enhance the activity, oxidation resistance, and flame retardancy of the material, among other properties [12,13]. Because the majority of plastic additives are not covalently bonded to the monomer polymer, they can be released into the environment and migrated [14]. Additionally, the additives used in plastic products have the potential to be harmful to organisms [15], but their environmental risks have been underestimated. The majority of plastic additives are endocrine disruptors, such as phthalates, brominated flame retardants, and bisphenol A [16,17,18,19], which are biotitic when released into the environment. Browne et al. [20] discovered that nonylphenol released by PVC significantly reduced Arenicolamarina’s coelomic osteocytes’ ability to remove pathogenic bacteria by more than 60%, and triclosan released by PVC significantly reduced its ability to produce deposits and resulted in worm death of more than 55%. Renzo et al. [21] carried out autopsies on 28 sea turtles from the Molise Adriatic coast. In all sea turtle fat and liver tissues, bisphenol A, terephthalic acid, and polyethylene terephthalate were detected. Nobre et al. [22] investigated the embryonic developmental toxicity of microplastic particles on *Lytechinus variegatus* and discovered that microplastic particles increased abnormal embryonic development levels by up to 66.5%, that microplastics can act as carriers of contaminants, and that plastic additives can also be leached into the environment and thus become toxic.

Because of the increasing usage of plastic products, disposing of this polymeric material has become a big concern for developed countries, as plastic pollution is persistent due to its durability [23,24,25]. Plastics are classified as thermosets and thermoplastics [26], with thermosets containing heteroatoms in the main chain, which are susceptible to hydrolytic degradation of chemical bonds such as ester or amide bonds, and thermoplastics containing long carbon chains in the main chain, which are resistant to hydrolytic degradation of chemical bonds [27]. As a thermoplastic, degradation of PS is even more challenging. Blanco et al. conducted a 13-month isothermal degradation experiment carried out at relatively low temperature (423 K) on PS and evidenced appreciable mass loss in the investigated period [28]. It was discovered that strain *Rhodococcus ruber* C208 could grow in the absence of a carbon source by using PS plastic as a carbon source [29]. After more than eight weeks of culture, strain C208 formed a dense biofilm and degraded a portion of PS [30], demonstrating that PS plastic can be disposed of via the biodegradation pathway.

Chen et al. [31] discovered that the aquatic toxicity of various plastic additive combinations varied and screened them for the best component combination of plastic components with the least aquatic toxicity. This paper employs keratinase from *Moesziomyces antarcticus* as a biodegradation protein for plastic PS and screens the major types of additives in plastic fractions that affect plastic degradation, including the best combination of plastic fractions that can be absorbed and degraded by microalgae to the greatest extent possible, in order to design a regulatory scheme for plastic biodegradation effect based on the optimization of additive fractionation ratio.

## 2. Materials and Methods

### 2.1. Biodegradability Assessment of the Plastic Component PS and Its Additives Using Molecular Docking and Molecular Dynamics Simulations

To investigate the biodegradability of plastic components (in this case, PS) and their additives by microorganisms, the PDB database (Protein Data Bank, PDB) was used to retrieve the keratinase (PDB ID: 7CC4) from a PS-degrading microorganism, *Moesziomyces antarcticus*, as a biodegrading protein for PS plastics [32].

Molecular docking is a technique for rapidly docking protein receptors with their corresponding ligand molecules. It is implemented in the DS software module Dock-Ligands (LibDock). It is primarily based on the Lock and Key principle of rigid binding between ligands and receptors, in which the interaction between ligands and receptors is modeled using the spatial shape and energy matching of the ligands and receptor macromolecules, and the complex’s conformation and binding affinity can be predicted [33]. Firstly, the incomplete amino acid residues of the protein were supplemented, and the protein was hydrogenated before molecular docking. Secondly, the Powell method was used to minimize the energy of molecules in SYBYL-2.0 software (parameters are as follows: Min Energy Change: 0.005 kcal/mol; Loading charge: Gasteiger-Huckel, Max Iterations: 10,000 Times; other parameters are default). All other parameters were set to their default values [34]. Finally, the module Receptor-Ligand Interactions is used to locate and define the receptor protein’s binding cavity, and the optimized ligand molecules are loaded into the DS software for docking with the receptor protein. The Libdock score is used to quantify the results of molecular docking, with higher scores indicating stronger interactions between ligand molecules and receptor proteins, more stable binding, and greater biodegradability [35].

Molecular Dynamics (MD) simulation is a molecular simulation technique based on Newtonian mechanics that is increasingly used in biochemistry, materials science, and other fields to obtain information about the thermodynamics and kinetics of receptor proteins and other macroscopic properties [36]. The dynamics of the molecular docking complex were simulated in this paper using GROMACS 4.6.5 software on a Dell PowerEdge R7525 server under the Gromos96 43A1 force field. At first, a cube box is constructed as the kinetic simulation region, into which the ligand-receptor complex is placed. Water is then added as the environmental medium in the blank group (the additive group must also add external stimuli) [37], and to balance the charge of the system and maintain it electrically neutral, Na^+^ is added to the box in the same amount as water molecules. Second, energy minimization via the fastest descent method, where the energy converges to 1000 kJ/mol and the system energy is equal to the equilibrium state’s energy. Then, in sequence, NVT temperature control (set at 300 K) and NPT pressure control (set to 1 bar to bind the ligand-protein complex site) are performed. Finally, at the MD equilibrium stage, the complex was unbound from its initial position and kinetically simulated at the initial set temperature and pressure using the frog-jump Newton integration method with a step length of 2 fs and 100,000 steps.

Additionally, the ligand’s binding energy (ΔG_b_) after docking with the acceptor (7CC4) was calculated using the MM-PBSA method (the binding energy is the sum of van der Waals energy, electrostatic energy, non-polar solvation energy, and polar solvation energy). In this article, a PS polymer (n = 5) was docked onto the 7CC4 receptor protein as a ligand-receptor complex structure for molecular dynamics (MD) simulations using 7CC4 as the receptor and blank set. Seven different types of plastic additives were added to the blank group as external stimuli and set up as an additive group [31]. These included common plasticizers, antioxidants, flame retardants, light stabilizers, heat stabilizers, lubricants, and fillers. The kinetics of the aforementioned blank and additive group complexes were simulated using the molecular mechanics Poisson-Boltzmann surface area (MM-PBSA) method in GROMACS 4.6.5 software using the Gromos96 43A1 force field, and the simulation results were expressed in terms of ΔG_b_ [38,39]. ΔG_b_ is used to quantify the extent to which a ligand binds to a receptor. The smaller the ΔG_b_, the more stable the complex formed between the ligand and the receptor [40]. The binding energy of PS polymers (n = 5) and their additive to additives to receptor proteins is used to characterize the biodegradability of microplastics in this article, with a lower ΔG_b_ indicating greater biodegradability.

### 2.2. Proportional Factorial Experimental Design Method for Screening the Proportioning Scheme of Plastic Components Affecting Biodegradability

Plastic, as a polymer material, is generally composed of polymer monomers and a variety of additives that vary according to the product and performance requirements. Common plastic products are primarily composed of polyolefin, polyvinyl chloride, and polyethylene terephthalate as raw materials, supplemented by plasticizers (A) (Diethylhexyl phthalate, Diisononyl phthalate, etc.), antioxidant (B) (Nonylphenol, Acetone diphenylamine, etc.), light stabilizers(C) (ultraviolet absorber UV-326, UV-328, etc.), Heat stabilizer (D) (Zinc stearate, Calcium stearate, etc.), Flame retardant (E) (commonly used brominated flame retardants: Decabromodiphenyl ether, Tetra bromo bisphenol A, etc.), Lubricant (F) (Oleamide, Stearic acid, etc.) and Filler (G) (Calcium carbonate, Calcium sulfate) and other mixtures of exogenous additives [41].The common polystyrene (PS)found in the environment is used as the plastic matrix in this paper, and seven typical plastic components (Table 1) are used as the primary research objects [31,42] to assess the biodegradability of microplastics in the environment. We adjusted the composition of plastic additives and screened for the presence of plastic components with a high potential for biodegradation.

Factorial design, or full factor experimental design, is a statistical method for obtaining a large amount of information and accurately estimating the size of each experimental factor’s main effect and interaction effect at all levels among factors. As a result, this method is capable of rapidly determining the optimal level of experimental factors [37]. With the assistance of Minitab 2021 software, and using the seven types of chemical additives in plastics (Plasticizer (A), Antioxidant (B), Light stabilizer (C), Heat stabilizer (D), Flame retardant (E), Lubricant (F) and Filler (G)) as the seven factors of factorial experimental design, each factor selects two representative substances as the level (Low level 1; High level 2).The “Statistics-DOE (Design of Experiment)-Factor-Create Factor Design” module generates a full factorial design experiment table of 128 (2^7^) plastic component combinations. Second, based on the optimal combination in Table 1, the six most critical additives (Plasticizer (A), Antioxidant (B), Light stabilizer (C), Heat stabilizer (D), Flame retardant (E), and Lubricant (F)) were used as six factors in the second round of factorial experimental design (Table 2), and each factor was added as a level (Low level 0, i.e., no addition; High level 1, i.e., addition), resulting64 (2^6^).

Additionally, based on the optimal additive composition formula (Table 1 No. 116), the response value was determined using biodegradability data quantified by dynamic binding energy. According to the factorial analysis method in the “Statistics-DOE (Design of Experiment)-Factor-Analysis Factor Design” module, which is capable of analyzing the interaction between various components of plastic additives and validating the underlying mechanism.

## 3. Results and Discussion

### 3.1. Biodegradability Features of Plastics Using the Entire Factorial Experimental Design Group’s Distribution Proportion Scheme

According to the method described in 2.2, 128 groups of various plastic additive components were created, and the binding free energy values of PS and receptor degradation protein 7CC4 in the blank (the blank group refers to the state in which no additives are present, and only polystyrene 5 polymer ligands and degraded protein receptors are present) and additive groups were calculated using a molecular docking-assisted molecular dynamics simulation method (Table 3), in order to characterize the size of plastic biodegradability. As shown in Table 3, except for the binding energies of the two combinations of No. 35 and No. 62, most of the binding energies decreased significantly when compared to the blank group (−100.374 kJ/mol) (114 groups in total). The binding energy values of the additive components were significantly reduced (at a rate of 20%), indicating that the addition and ratio of additives aided in the biodegradation of PS plastics, which could significantly improve the biodegradability of microplastics, and also demonstrate the necessity of adding exogenous additives during the plastic manufacturing process [44].

Additionally, the binding energy of additive group No. 116 (−192.483 kJ/mol) is the lowest of all additive combinations and decreases by the greatest amount (91.77%) (The percentage of change rate refers to the change rate of binding energy in the presence of additives compared to the blank group (that is, the change rate of degradability = (binding energy value of additive group—binding energy value of blank group)/binding energy value of blank group × 100%)) when compared to the binding energy of the blank group (−100.374 kJ/mol), indicating the additive combination (Diisononyl phthalate, Acetone diphenylamine, Bometriazole, Zinc stearate, Tetrabromobisphenol A, Oleamide and Calcium sulfate). Thus, using combination No. 116 as an example, the following analysis was conducted to determine the effect of various additive components and ratios on the biodegradability of plastics.

### 3.2. Analysis of the Primary Impacts of Plastic Additives and Their Interactions on the Biodegradability of Plastics

#### 3.2.1. Biodegradation Characteristics of Plastics under an Additive Combination Scheme Using a Full Factorial Design

Using a fractional factorial experimental design, the effects of plastic additives on the biodegradability of plastics were studied using the optimal biodegradability combination scheme No. 116 of polymers screened in Section 3.1.

According to Table 4, the binding energies of antioxidant (B) (Acetone diphenylamine), heat stabilizer (D) (Zinc stearate) and filler G (Calcium sulfate) are −146.232 kJ/mol, −143.404 kJ/mol and −144.624 kJ/mol, respectively. The decreased range is significantly less than that of the other four additives, indicating that these three additives have a less favorable effect on plastic’s biodegradability than the other four (Plasticizer (A): Isopropyl phthalate, Light stabilizer (C): Bumetrazole, Flame retardant (E): Tetrabromobisphenol A and Lubricant (F): Oleamide). Additionally, filler (Calcium sulfate) has a smaller part in plastic production than other additives, and it is mostly employed to minimize manufacturing costs [45]. Thus, to more effectively analyze the effect of plastic additives on plastic biodegradability, six different types of plastic additives were chosen (A, B, C, D, E, and F), and a full factorial experimental design of six factors and two levels was used to generate 64 groups of plastic additive combination schemes (Table 5), in order to identify the main effects of plastic additives and their interaction effects on plastic biodegradability.

#### 3.2.2. The Effect of Plastic Additives on the Biodegradability of Polymers Was Investigated

The contribution of plastic additives to polymer biodegradability may not be completely independent. The DOE-analysis factor design module of Minitab software was used in this work to determine the major influence and the kind and degree of the interaction effect of additive components on the biodegradability of plastics. The primary effects of plastic additives and their interaction effects on plastic degradation were investigated, as well as the contribution rate of each order interaction effect. The results are summarized in Table 6 and Figure 1.

Among the interactions of plastic additive components contributing to the biodegradability of plastics, the third-order interaction accounts for the highest proportion, followed by the second-order interaction effect, indicating that the interaction between the components of plastic additives is complex, and multi-factor synergistic or antagonistic effects dominate.

Among the major effects, plasticizer (A) has the greatest synergistic effect when microplastics and degradation proteins are combined, while antioxidant (B) has the greatest inhibitory effect when microplastics and degradation proteins are combined. In the second-order interaction effect, the coexistence of plasticizer (A) and lubricant (F) is most favorable for the combination of microplastics and degradation proteins, followed by the synergistic effect of plasticizer (A) and antioxidant (B) (this paper uses 10% as the significant standard); the combination of light stabilizer (C) and flame retardant (E) has the most significant antagonistic effect on the combination of microplastics and degradation proteins. Additionally, antioxidant (B) and heat stabilizers (D) work in opposition to the combination of plastic additives and breakdown proteins.

The effect is strong, and its magnitude exceeds 20%. As a result, it is not recommended to employ antioxidant (B) and heat stabilizer (D) as plastic additives concurrently to avoid complicating plastic biodegradation. In the third-order interaction effect, when antioxidant (B), light stabilizer (C), and heat stabilizer (D) are combined; the combination of plastic additives and degradation proteins is considerably increased, as is the combination of antioxidant (B), flame retardant (E) and lubricant (F) can be significantly increased.

Additionally, the combination of plasticizer (A), light stabilizer (C), and flame retardant (E) has a considerable synergistic impact when plastic additives and degradation proteins are combined. The combination of plasticizer (A), antioxidant (B), and flame retardant (E) is, in some ways, incompatible with the combination of plastic additives and breakdown proteins, but the effect is negligible.

The DOE module in Minitab was used to check the consistency of the degradation effect data for 64 groups of microplastics ((C_8_H_8_)_n, n = 5_) additive combinations. Figure 2 depicts the results. All the schemes in this work follow the normal distribution trend, as seen in Figure 2. The residual distribution histogram trend is to increase first, then decline. The residual plots are dotted, indicating that the results are accurate. There is no aberrant change trend. The results are not related to the experimental order in the graph of residuals and experimental order, indicating that the experimental order has no effects on the results, demonstrating that the results are independent.

### 3.3. Factorial Analysis and Molecular Dynamics Simulation Were Used to Investigate the Mechanism of Plastic Biodegradation

#### 3.3.1. Molecular Dynamics Simulation-Based Mechanism Investigation of the Biodegradation Effect of Polystyrene 5 Polymers

The dominance analysis is primarily a statistical tool for determining the interdependence of multiple variables. As a result, this work employs the dominance analysis method to investigate the primary influencing elements of 64 different plastic additive combinations on the degradation of the polystyrene 5 polymer via protein degradation (7CC4). The van der Waals energy, electrostatic energy, polar solvation energy and non-polar solvation energy make up the binding free energy of ligand-receptor docking [46]. The lower the free energy, the better the biodegradability of the polystyrene 5 polymer. The correlation between the components of binding free energy of the polystyrene 5 polymer and receptor 7CC4 and the change value of binding free energy was analyzed under 64 additive combinations, with no external additive combination (No. 116–16) as the invariant, and which energies have a significant correlation with the binding free energy was analyzed.

Figure 3 shows the 95% confidence interval in grey, with different colors representing distinct components: the bigger the correlation coefficient r, the stronger the association [47]. The correlation coefficients r of the linear regression equations between the change values of the combined free energy (the difference compared to the blank group No. 116–16) of the 64 combination schemes and van der Waals energy, non-polar solvation energy, polar solvation energy, and electrostatic energy, respectively, were 0.8185, 0.7483, 0.1010 and 0.0608 (r_α_ = 0.2461, *p* = 0.05, n = 63, r > r_α_ was significantly correlated). Furthermore, the change range of van der Waals energy is far wider than the jump range of non-polar solvation energy (SASA energy) under different additive combinations, both of which are negative, but the latter is smaller in absolute value. The solvation energy of the system is all positive, and its absolute value is larger than the absolute value of the SASA energy, indicating that the solvation effect is not conducive to the combination of the polystyrene 5 polymer and biodegrading enzymes, and the van der Waals energy is certain. Furthermore, there is a significant beneficial effect on the biodegradability of polystyrene 5 polymer. In addition, it has been found that the binding cavity of proteases is hydrophobic [47,48]. Moreover, combined with the factorial analysis results in Section 3.2, it can be found that the chemical structure of the additive with significant promoting effect mainly exhibits hydrophobic properties. Therefore, when the additive is used as an external stimulus, the additive will promote the further inward folding of the degraded protein, so that the binding of the degraded protein to the polystyrene pentamer is enhanced.

#### 3.3.2. Total Factor and Amino Acid Residue Analysis Were Used to Investigate the Mechanism of Biodegradation of Polystyrene 5 Polymers

The findings of factorial analysis, when combined with Figure 1 and Figure 4, revealed that the effect of interaction between different additives on the biodegradation of the polystyrene 5 polymer was distinct. The absolute magnitude of the biodegradation standardization effect of the polystyrene 5 polymer is depicted in the Pareto diagram (Figure 4). (From the maximum to the minimum effect). It is statistically significant (α = 0.05) when the factor bar passes the reference line 2.074. As a result, the impact of additive A and its combination B*C*D, C*E, B*E*F, and A*C*E on the polystyrene 5 polymer’s biodegrading performance was statistically significant (Figure 4a), and the addition of B*C*D, A, B*E*F, and A*C*E has a substantial favorable effect. That is, it promotes the breakdown of the polystyrene 5 polymer (Figure 4b).

The amino acid residues analysis of the interaction between the polystyrene 5 polymer and degradation enzyme 7CC4 is displayed in Figure 5 under the condition of an external addition with a clear favorable effect (where the green circle represented the amino acid residues involved in van der Waals interaction [49]).The number of amino acid residues around the polystyrene 5 polymer increased after adding external conditions conducive to polystyrene 5 degradation, indicating that van der Waals interaction was the main driving force for the binding of the polystyrene 5 polymer and degradation protein 7CC4. This conclusion follows the preceding one. Hydrogen bonds and hydrophobic interactions are critical in the binding of ligands to proteases, according to previous research [50]. Table 7 shows that in the absence of external additives (No. 116–16), only four hydrophobic interactions were formed between the polystyrene 5 polymer and degradation protein 7CC4, whereas a combination of external additives (No. 116–42, No. 116–28, No. 116–30, No. 116–45) formed seven, five, five and eight hydrophobic interactions, respectively, with only a small amount of hydrogen bonds, indicating that the more hydrophobic interactions formed, the more conducive to the combination of the polystyrene 5 polymer and degradation protein 7CC4. As a result, the reason the degradation effect of polystyrene 5 polymers is stronger in the group with favorable external conditions than in the No. 116–16 group may be because the former generated more hydrophobic contacts. Combined with the factorial analysis results in Section 3.2, the No. 116–45 group contains the plasticizer with the highest proportion of plastic additives, and the factorial result of this additive combination shows a significant positive effect; the polystyrene 5 polymer formed more hydrophobic interactions with the degradation protein 7CC4, compared to the other three positive effect groups. As a result, the No. 116–45 group was recommended as the best combination scheme for the biodegradation of polystyrene 5 polymers in the study.

#### 3.3.3. Based on Factorial Design Verification, We Examined the Biodegradation Mechanism of the Polystyrene 5 Polymer

Zhang et al. [51] developed five environmentally friendly PAEs derivatives (DBP–CHO, DBP–COOH, DINP–NH_2_, DINP–NO_2_) using a normalization method, a pharmacophore model, and molecular docking technology. They discovered that DBP–CHO had a high biodegradability, was less toxic, and more easily degraded by light in the natural environment.

To further validate the rationality and universality of the design theory and factorial experiment design used in this study, the additive combinations with plasticizer A with an effective contribution percentage greater than 10% in the factorial results were screened and verified. The plasticizers (A) in the groups (No. 116–28, No. 116–40, No. 116–26, No. 116–45) were replaced with the above five derivatives, and the replaced plasticizer was used as a new external condition (Table 8).

As indicated in Figure 1, all four combinations had a positive effect value, and the contribution percentage was as follows: No. 116–28 > No. 116–26 > No. 116–45 ≈ No. 116–40. The analysis and calculation of the binding free energy data revealed that only after the DBP–CHO derivative molecule was replaced by A was the change trend of the binding free energy of the polystyrene 5 polymer and 7CC4 protease in the four groups consistent with the change trend in the contribution percentage of the effect value (the binding free energy values of the four groups were greater than those of No. 116–16 group). The screening results were consistent with those of Zhang et al., indicating that DBP–CHO had a favorable effect on the biodegradability of polystyrene 5, indicating the rationality, universality, and reliability of the design theory, the factorial experiment design, and the factorial analysis results in this study.

## 4. Conclusions

In this research, a biodegradation effect regulatory system based on the optimization of the number of plastic additives was created using factorial design and associated analysis methods supplemented by molecular docking and molecular dynamics simulation approaches. The investigation of relevant mechanisms revealed that van der Waals contact was highly associated with plastic’s biodegradability. Simultaneously, it was discovered that hydrophobic interaction was the primary driving force behind the biodegradation of plastics, guiding the efficient biodegradation of plastics under the condition of a reasonable additive component combination.

## Figures and Tables

**Figure 1 ijerph-19-05670-f001:**
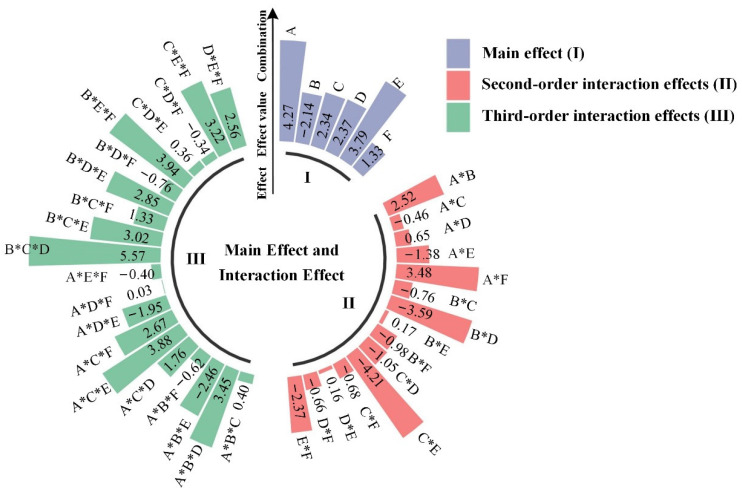
Main effects, second order and third-order interaction effects of polystyrene 5- polymer and degradation protein 7CC4 in the coexistence of plastic additives. (* Implies simultaneous use of multiple components).

**Figure 2 ijerph-19-05670-f002:**
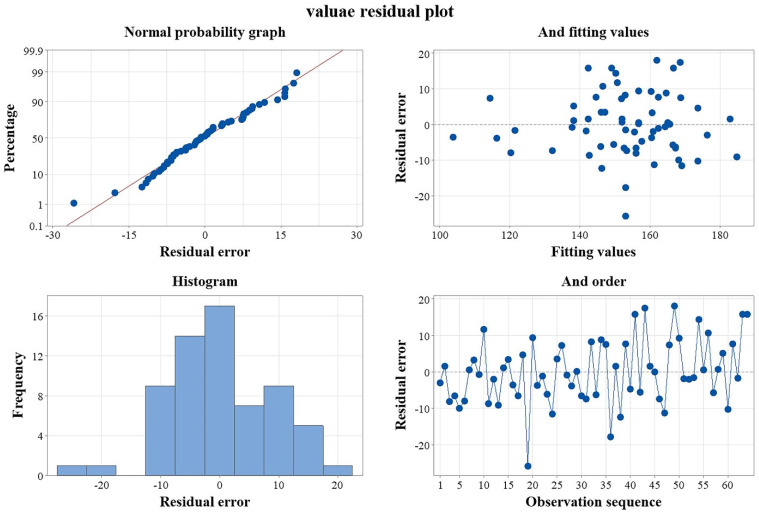
Reliability test of factorial analysis between polystyrene 5 polymer and biodegraded protein under the combination scheme of plastic additives.

**Figure 3 ijerph-19-05670-f003:**
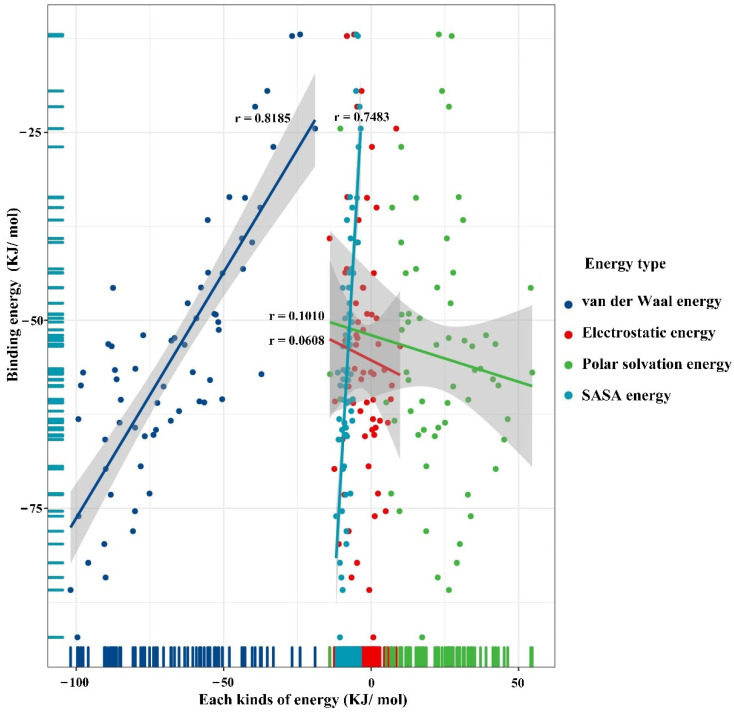
Linear regression investigation of the binding free energy of polystyrene 5 polymers.

**Figure 4 ijerph-19-05670-f004:**
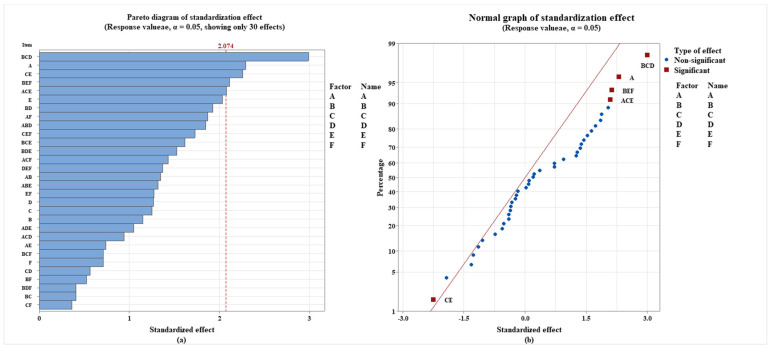
Analysis of the degree of main effect and interaction effect of plastic additives on polystyrene 5 polymer biodegradation (Pareto Diagram of standardization effect (**a**), normal diagram of standardization effect (**b**)).

**Figure 5 ijerph-19-05670-f005:**
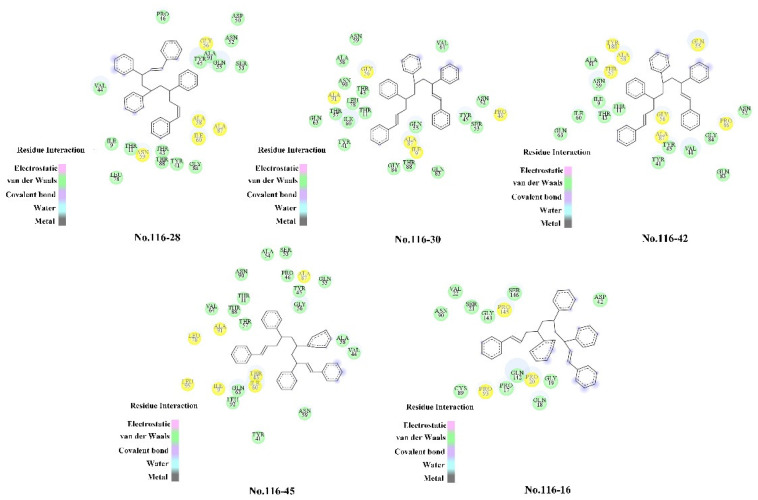
Amino acid residue diagram of polystyrene 5 polymer interacting with degrading enzyme (7CC4) under the condition of external additive with obvious positive effect. (After molecular dynamics simulation, the complex of ligand and degrading enzyme (7CC4) was extracted, and the interaction between ligand and degrading enzyme was analyzed. The yellow circle represented the main amino acid residues in the interaction between degrading enzyme (7CC4) and polystyrene pentamer).

**Table 1 ijerph-19-05670-t001:** Typical polymeric additives and their maximum and minimum concentrations.

Compositions	Chemicals	Molecular Formula	CAS Number
Plastic Matrix	Polystyrene	(C_8_H_8_)_n_, _n = 5_	100-42-5
Plasticizers (A)	Diethylhexyl phthalate (1)	C_24_H_38_O_4_	117-81-7
	Diisononyl phthalate (2)	C_26_H_42_O_4_	28553-12-0
Antioxidants (B)	Acetone diphenylamine (1)	C_15_H_17_NO	68412-48-6
	Nonyl phenol (2)	C_15_H_24_O	25154-52-3
Light stabilizers (C)	2-(2H-benzotriazol-2-yl)-4,6-di-tert-pentylphenol (1)	C_22_H_29_N_3_O	25973-55-1
	Bumetrizole (2)	C_17_H_18_N_3_OCl	3896-11-5
Heat stabilizers (D)	Calcium stearate (1)	C_36_H_70_O_4_Ca	1592-23-0
	Zinc stearate (2)	C_36_H_70_O_4_Zn	557-05-1
Flame retardants (E)	Decabromodiphenyl ether (1)	C_12_Br_10_O	1163-19-5
	Tetrabromobisphenol A (2)	C_15_H_12_Br_4_O_2_	79-94-7
Lubricants (F)	Oleamide (1)	C_18_H_35_NO	301-02-0
	Stearic acid (2)	C_18_H_36_O_2_	57-11-4
Fillers (G)	Calcium carbonate (1)	CaCO_3_	471-34-1
	Calcium sulfate (2)	CaSO_4_	10101-41-4

Note: 1: Low level; 2: High level; 7 types of exogenous additives according to A:B:C:D:E:F:G = 15%:3%:3%:3%:3%:1.5%:1.5% (converted to the number of A:B:C:D:E:F = 10:2:2:2:2:1:1) add [43].

**Table 2 ijerph-19-05670-t002:** Setting the proportioning system for plastic additives.

Compositions	Chemicals	Molecular Formula	CAS Number
Plastic Matrix	Polystyrene	(C_8_H_8_)_n_, _n = 5_	100-42-5
Plasticizers (A)	None (0)	-	-
	Diisononyl phthalate (1)	C_26_H_42_O_4_	28553-12-0
Antioxidants (B)	None (0)	-	-
	Acetone diphenylamine (1)	C_15_H_17_NO	68412-48-6
Light stabilizers (C)	None (0)	-	-
	Bumetrizole (1)	C_17_H_18_N_3_OCl	3896-11-5
Heat stabilizers (D)	None (0)	-	-
	Zinc stearate (1)	C_36_H_70_O_4_Zn	557-05-1
Flame retardants (E)	None (0)	-	-
	Tetrabromobisphenol A (1)	C_15_H_12_Br_4_O_2_	79-94-7
Lubricants (F)	None (0)	-	-
	Oleamide (1)	C_18_H_35_NO	301-02-0

Note: 0: Low level; 1: High level; 6 types of exogenous additives according to A:B:C:D:E:F:G = 15%:3%:3%:3%:3%:3% (converted to the number of A:B:C:D:E:F = 10:2:2:2:2:2) add [43].

**Table 3 ijerph-19-05670-t003:** The proportioning strategy for plastic components is based on a full factorial experimental design and the binding energies of the components with breakdown proteins (7CC4).

No.	Combination	Binding Energy (kJ/mol)	Change Rate (%)	No.	Combination	Binding Energy (kJ/mol)	Change Rate (%)
Blank Group	−	−100.374	−	Blank Group	−	−100.374	−
**1**	2122121	−165.347	64.73	**65**	1211212	−127.231	26.76
**2**	1212111	−159.899	59.30	**66**	2121112	−139.953	39.43
**3**	2222111	−171.493	70.85	**67**	1222212	−146.936	46.39
**4**	2211222	−149.510	48.95	**68**	1121221	−110.175	9.76
**5**	1212121	−149.627	49.07	**69**	2222212	−181.704	81.03
**6**	2111211	−176.630	75.97	**70**	2211112	−149.459	48.90
**7**	1212211	−189.748	89.04	**71**	2222112	−182.239	81.56
**8**	1122122	−115.771	15.34	**72**	2222222	−188.257	87.56
**9**	1211121	−165.639	65.02	**73**	2222211	−147.114	46.57
**10**	2211122	−116.508	16.07	**74**	2122222	−118.484	18.04
**11**	2122211	−167.863	67.24	**75**	1211111	−129.242	28.76
**12**	2211211	−168.668	68.04	**76**	2212121	−101.028	0.65
**13**	2211212	−139.039	38.52	**77**	2211121	−139.678	39.16
**14**	2111122	−133.583	33.09	**78**	1222121	−135.564	35.06
**15**	1111122	−146.259	45.71	**79**	2121121	−166.372	65.75
**16**	1121222	−142.824	42.29	**80**	1222211	−168.897	68.27
**17**	2221221	−149.945	49.39	**81**	1222222	−179.741	79.07
**18**	1111211	−186.012	85.32	**82**	2121122	−139.483	38.96
**19**	2111212	−140.396	39.87	**83**	1212122	−153.210	52.64
**20**	1211222	−122.226	21.77	**84**	1221212	−174.317	73.67
**21**	1111112	−159.221	58.63	**85**	1122211	−175.672	75.02
**22**	2211221	−176.404	75.75	**86**	2122221	−164.832	64.22
**23**	1212221	−162.598	61.99	**87**	1121122	−143.276	42.74
**24**	1221112	−133.649	33.15	**88**	1111111	−168.066	67.44
**25**	1211211	−146.415	45.87	**89**	2212211	−145.248	44.71
**26**	2111222	−104.090	3.70	**90**	2121212	−108.537	8.13
**27**	1121212	−162.534	61.93	**91**	1222111	−126.167	25.70
**28**	1121211	−124.222	23.76	**92**	1122111	−162.668	62.06
**29**	2211111	−185.857	85.16	**93**	1112211	−156.718	56.13
**30**	2112211	−151.042	50.48	**94**	1111222	−130.514	30.03
**31**	2122111	−156.230	55.65	**95**	1212222	−185.417	84.73
**32**	2112112	−144.408	43.87	**96**	1112122	−172.772	72.13
**33**	1122212	−171.231	70.59	**97**	2111221	−153.231	52.66
**34**	1111212	−160.521	59.92	**98**	1112212	−137.189	36.68
**35**	1222221	−80.943	−19.36	**99**	2212111	−173.947	73.30
**36**	1221222	−133.329	32.83	**100**	1212212	−166.364	65.74
**37**	1222122	−126.188	25.72	**101**	1112111	−179.134	78.47
**38**	2112122	−143.624	43.09	**102**	2221112	−160.316	59.72
**39**	2122122	−174.420	73.77	**103**	1121111	−156.163	55.58
**40**	2111121	−117.572	17.13	**104**	1211112	−162.505	61.90
**41**	2111111	−176.795	76.14	**105**	2222122	−133.804	33.31
**42**	2121222	−149.533	48.98	**106**	1121121	−135.199	34.70
**43**	1222112	−163.450	62.84	**107**	1122112	−187.302	86.60
**44**	2212122	−140.908	40.38	**108**	2122112	−122.819	22.36
**45**	2221212	−143.580	43.05	**109**	2222221	−101.376	1.00
**46**	1221221	−129.625	29.14	**110**	1122221	−169.289	68.66
**47**	2121211	−175.918	75.26	**111**	2221122	−172.425	71.78
**48**	2212221	−172.734	72.09	**112**	2222121	−157.271	56.68
**49**	2112212	−163.466	62.86	**113**	2112111	−159.225	58.63
**50**	1221121	−171.508	70.87	**114**	1212112	−151.288	50.72
**51**	1112221	−180.381	79.71	**115**	2212222	−175.982	75.33
**52**	2112221	−164.344	63.73	**116**	2122212	−192.483	91.77
**53**	2212212	−155.909	55.33	**117**	2212112	−163.769	63.16
**54**	1211122	−157.855	57.27	**118**	1122222	−158.070	57.48
**55**	2112121	−174.158	73.51	**119**	1221211	−160.681	60.08
**56**	1111221	−120.981	20.53	**120**	1121112	−170.986	70.35
**57**	1112121	−151.498	50.93	**121**	1112222	−124.770	24.31
**58**	2221121	−186.537	85.84	**122**	2112222	−163.022	62.41
**59**	2221111	−106.280	5.88	**123**	1112112	−138.551	38.03
**60**	1111121	−104.089	3.70	**124**	1211221	−159.955	59.36
**61**	1122121	−132.955	32.46	**125**	1221111	−168.355	67.73
**62**	2221222	−84.160	−16.15	**126**	2111112	−104.542	4.15
**63**	2221211	−172.479	71.84	**127**	2121221	−159.279	58.69
**64**	2121111	−151.320	50.76	**128**	1221122	−161.112	60.51

Note: 1 indicates a low level; 2 indicates a high level.

**Table 4 ijerph-19-05670-t004:** Binding energy of Polystyrene 5 polymer and additives to plastic in the presence of breakdown protein (7CC4).

Plastic Composition	Binding Energy (kJ/mol)
(C_8_H_8_)_n_, _n = 5_	−100.374
Diisononyl phthalate	−183.324
Acetone diphenylamine	−146.232
Bumetrizole	−153.988
Zinc stearate	−143.404
Tetrabromobisphenol A	−164.449
Oleamide	−186.741
Calcium sulfate	−144.624

**Table 5 ijerph-19-05670-t005:** A plan for combining plastic additives based on six components, two levels, and a full factorial design, as well as a description of its biodegradation impact.

No.	Combination	Binding Energy (kJ/mol)	Change Rate (%)	No.	Combination	Binding Energy (kJ/mol)	Change Rate (%)
**No. 116-1**	000111	−173.410	72.76	**No. 116-33**	100010	−161.165	60.56
**No. 116-2**	000110	−184.535	83.85	**No. 116-34**	001001	−173.560	72.91
**No. 116-3**	001110	−148.113	47.56	**No. 116-35**	110001	−176.420	75.76
**No. 116-4**	010110	−146.021	45.48	**No. 116-36**	110011	−135.375	34.87
**No. 116-5**	101011	−158.323	57.73	**No. 116-37**	010111	−153.645	53.07
**No. 116-6**	010100	−112.337	11.92	**No. 116-38**	011010	−133.981	33.48
**No. 116-7**	100111	−165.617	65.00	**No. 116-39**	000100	−152.368	51.80
**No. 116-8**	001010	−163.996	63.38	**No. 116-40**	110000	−153.090	52.52
**No. 116-9**	000101	−163.734	63.12	**No. 116-41**	101000	−158.225	57.64
**No. 116-10**	110111	−162.456	61.85	**No. 116-42**	011100	−144.108	43.57
**No. 116-11**	111000	−134.082	33.58	**No. 116-43**	101101	−186.196	85.50
**No. 116-12**	101111	−159.050	58.46	**No. 116-44**	001111	−144.039	43.50
**No. 116-13**	111111	−175.752	75.10	**No. 116-45**	101010	−165.794	65.18
**No. 116-14**	010000	−139.483	38.96	**No. 116-46**	110101	−146.067	45.52
**No. 116-15**	010001	−150.620	50.06	**No. 116-47**	111110	−150.120	49.56
**No. 116-16**	000000	−100.374	-	**No. 116-48**	010101	−121.953	21.50
**No. 116-17**	100110	−160.875	60.28	**No. 116-49**	111011	−180.120	79.45
**No. 116-18**	001100	−178.409	77.74	**No. 116-50**	110010	−169.788	69.16
**No. 116-19**	101100	−127.301	26.83	**No. 116-51**	111010	−140.016	39.49
**No. 116-20**	011000	−166.234	65.61	**No. 116-52**	111001	−153.551	52.98
**No. 116-21**	001101	−156.825	56.24	**No. 116-53**	011111	−151.653	51.09
**No. 116-22**	111101	−161.370	60.77	**No. 116-54**	011110	−164.654	64.04
**No. 116-23**	011001	−139.879	39.36	**No. 116-55**	110100	−157.329	56.74
**No. 116-24**	101001	−157.547	56.96	**No. 116-56**	100100	−157.288	56.70
**No. 116-25**	011011	−149.624	49.07	**No. 116-57**	001000	−160.944	60.34
**No. 116-26**	100001	−159.186	58.59	**No. 116-58**	110110	−152.739	52.17
**No. 116-27**	001011	−137.040	36.53	**No. 116-59**	000011	−143.545	43.01
**No. 116-28**	100000	−112.544	12.12	**No. 116-60**	100101	−163.517	62.91
**No. 116-29**	000010	−156.999	56.41	**No. 116-61**	010010	−170.14	69.51
**No. 116-30**	010011	−149.525	48.97	**No. 116-62**	011101	−119.86	19.41
**No. 116-31**	000001	−124.866	24.40	**No. 116-63**	101110	−164.943	64.33
**No. 116-32**	100011	−161.298	60.70	**No. 116-64**	111100	−182.603	81.92

Note: 0: Low level; 1: High level.

**Table 6 ijerph-19-05670-t006:** The major effects of plastic additives on the total contribution rate of plastic biodegradability, as well as their interactions.

Contribution Rate	Percentage (%)
Main effect	20.68
Second-order interaction effects	25.16
third-order interaction effects	54.16
Total	100

**Table 7 ijerph-19-05670-t007:** Significantly positive non-bond interactions between polystyrene 5 polymers and 7CC4 in the presence of exogenous additives.

Groups	Combination	No Bonded Interaction	Interaction Amino Acids	Number
No. 116–16	None	Mixed Pi/Alkyl Hydrophobic	PRO145, PRO93, PRO20	4
No. 116–42	B*C*D	Pi-Alkyl Hydrophobic Mixed Pi/Alkyl Hydrophobic	ALA87, PRO46, GLN55, GLY56, THR57, ALA58, TYR180	3 4
No. 116–28	A	Hydrogen Bonds (no classical) Pi-Alkyl Hydrophobic	GLY56, ALA58, ALA87, ILE60, ASN59	1
				5
No. 116–30	B*E*F	Hydrogen Bonds Mixed Pi/Alkyl Hydrophobic	GLY56, ALA91, ALA87, ILE9, PRO46	1
				5
No. 116–45	A*C*E	Mixed Pi/Alkyl Hydrophobic	ALA87, ALA91, LEU78, LEU95, ILE9, THR43, ILE60	8

Note: The data in the table were extracted from polystyrene 5 polymers and 7CC4 protein complex under MD simulation equilibrium. * Implies simultaneous use of multiple components.

**Table 8 ijerph-19-05670-t008:** The binding free energy of polystyrene 5 polymers with 7CC4 degrading protein after substituting PAE derivatives for A (plasticizer).

Groups	PAE Substitutes									
	DBP-CHO		DBP-COOH		DBP-OH		DINP-NH_2_		DINP-NO_2_	
	${\Delta G}_{b}(\mathrm{kJ}/\mathrm{mol})$	Change Rate (%)	${\Delta G}_{b}(\mathrm{kJ}/\mathrm{mol})$	Change Rate (%)	${\Delta G}_{b}(\mathrm{kJ}/\mathrm{mol})$	Change Rate (%)	${\Delta G}_{b}(\mathrm{kJ}/\mathrm{mol})$	Change Rate (%)	${\Delta G}_{b}(\mathrm{kJ}/\mathrm{mol})$	Change Rate (%)
**No. 116–28** (A)	−181.290	80.61%	−123.126	22.67%	−132.944	32.45%	−151.873	51.31%	−167.071	66.45%
**No. 116–40** (A*B)	−159.726	59.13%	−157.740	57.15%	−175.469	74.82%	−139.738	39.22%	−156.185	55.60%
**No. 116–26** (A*F)	−186.207	85.51%	−164.083	63.47%	−166.870	66.25%	−155.496	54.92%	−146.213	45.67%
**No. 116–45** (A*C*E)	−153.546	52.97%	−138.699	38.18%	−163.177	62.57%	−134.371	33.87%	−161.306	60.70%

* Implies simultaneous use of multiple components.

## Data Availability

Not applicable.

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
