# Peer review of "Control Strategies of Plastic Biodegradation through Adjusting Additives Ratios Using In Silico Approaches Associated with Proportional Factorial Experimental Design"

_ijerph, 2022, doi:10.3390/ijerph19095670_

Round 1

Reviewer 1 Report

The manuscript by these Authors is a very interesting study as regards the control of the plastic biodegradation by regulating the component ratios. The topic is actual and the impact of plastic, in particular its persistence, is a key sustainability point. The manuscript is well written, despite the quality of the presentation can be improved (i.e. improve the graphic presentation). I would suggest stressing, in the introduction, the point regarding the plastic persistence for a long time in the environment (see attached .pdf for specific suggestions. I would also suggest a slight revision of the text to eliminate mistakes and typos (see attached .pdf). I strongly recommend the publication of this paper.

Reviewer 2 Report

Dear authors

You've made a very good work with a lot of calculations and nice hypotheses to be assessed. However, I have some comments and recommendations to do:

  • In a more general approach: Along the paper you specify " polymer degradation" when really the word should be "biodegradation" . All the planning with software about docking molecules is referred to proteínes what is only possible in biodegradatioand not chemical degradation.
  • The description of blank molecules used should be extended because is very difficult to see where the differences on energy come from is not specified or described in some sense
  • The results obtained would be interesting to be presented separating the different types of molecular interactions considered. If not, conclusions are evident when assuming that the docking process will be more probable in the more organic groups what, inmediately means van der waals interactions. It is not necessary to use software and statistical planning for to detect that fact.
  • It should be, clearly specified which are the type of interactions between polymer and additives during the docking process. From the information delivered in the paper, it seems that interactions are studiend isolated from the polymer. Please, consdier to clarify everything related with this point.

In more specific aspects: 

Authors 2 and 3 show the same e-mail

-Page 2: Please, consider to add the size of the microplastics consdiered as reference because that part is also very important for readers to better understand the meaning of the parameter n, in the simulation. From Fig 2 the length of the chain is crucial when dealing with the dynamics of molecules

-Page 2, line 87. Please consdier to describe C208, or at least, indicate the microorganisms used. When consulting bibliography, with the same denomination (Rhodoccus Rubber), there are several other enzymes as laccase coming from this specific strain, when you talk about keratinase (page 3, line 104). Please, consider to clarify this aspect 

  • Page 3, lines 111 to 114. Please, consdier to revise the text used,
  • Line 136. Please, specify the meaning of NVT and NPT
  • Line147. The explanation of how blank group is choosen, keeps very confusing for readers. Please, consdier to extent the explanation about the procedure used
  • page 4, line 155. When trying to justify the use of n=5, it keeps not very clear how do you decide to choose it. Free Gibbs energy of the binding energy can give to the formation of a more stable complex without reaction of degradation
  • page 6, line209 and 213. Please, consdier to rextent the explanation of the decission about 35 and 62, specially when the definition of the blank group is not clearly explained , as well as the Change rate percentage
  • Table 3, please consider to decrease the size of letters to fit into the wiidth of the column 
  • Afeter paragraph 3.2, maybe it would be interesting to add a figure with picture explaining the process followed to use the blank group and the comparison made
  • Table 4. Binding energy of what?? Between them, between the additives with the polymer consdiered as n=5 or the general chains of polymer?? 
  • Legend of Table 5: check if the numer "six components" is OK, in my opinion we are talking about seven components, excluding the base chain of polymer
  • Paragraph 3.2.2. Lines 253 to 254 revise the phrase used or explain the concept independent when dealing to a chemical compounds totally dispersed into the mass of plastic away enough for not to experience interactions between them and only with the PS
  • Page 10. During the explanation in Lines 265 and 269, it keeps not clear if you're considering the interactions with polymer
  • Page 12. Please, consider to change the legend of Fig 1. In my opinion it is not describing the statistical parameters shown there
  • Fig 2. Should be improved and x-axis specified clearly. Text is not possible to red 
  • Page 13. Lines 329 to 331. Please, consider to extend the explanation because this affirmation would be had done just attending the chemical structure of additives and the organic character of the proteins used
  • Page 14, Lines 354 to 359. Why not to summarize all these explanations using a Table with the detailed information about the type of interactions detected and their relative importance??
  • Page 15. Fig 4. This picture should be improved and better explained
  • Table 9. Is the percantage of change the same that was used in anterior tables?? Or there are different molecules used as reference. it keeps not clear 

Many thanks

Reviewer 3 Report

This is an interesting work about the use of molecular dynamics simulation for the prediction of plastics biodegradation  in the presence of different types of additives.  The subject of the paper is of good interest, and the use of statistics allows to identify the main factors which can promote a faster degradation of the PS used as model polymer.

I have only one observation to the reported results, which is however very important. The authors have used a PS with a degree of polymerization n=5. This is a very low value, of course, which does not correspond to a commercial polymer. Do the authors expect that the value of n can play a significant role in determining the biodegradation of PS, also in the presence of other additives. I think can a preliminary estimation of the effect of n could strengthen the paper and related results.

Round 2

Reviewer 2 Report

Dear authors

Many thanks for your efforts in the revision process. You have taken into consideration all my comments and, In my opinion, the article is now ready to be published